# Metabolomics-Driven Discovery of an Introduced Species and Two Malaysian *Piper betle* L. Variants

**DOI:** 10.3390/plants10112510

**Published:** 2021-11-19

**Authors:** Muhamad Faris Osman, Soo Yee Lee, Shahrul Razid Sarbini, Siti Munirah Mohd Faudzi, Shamsul Khamis, Badrul Hisyam Zainudin, Khozirah Shaari

**Affiliations:** 1Natural Medicines and Products Research Laboratory (NaturMeds), Institute of Bioscience, Universiti Putra Malaysia, UPM, Serdang 43400, Selangor, Malaysia; farisosman@iium.edu.my (M.F.O.); daphne.leesooyee@gmail.com (S.Y.L.); sitimunirah@upm.edu.my (S.M.M.F.); 2Department of Pharmaceutical Chemistry, Kulliyyah of Pharmacy, International Islamic University Malaysia, Kuantan 25200, Pahang, Malaysia; 3Department of Crop Science, Faculty of Agricultural Science and Forestry, Universiti Putra Malaysia, Bintulu 97008, Sarawak, Malaysia; shahrulrazid@upm.edu.my; 4Department of Chemistry, Faculty of Science, Universiti Putra Malaysia, UPM, Serdang 43400, Selangor, Malaysia; 5Department of Biological Sciences and Biotechnology, Faculty of Science and Technology, Universiti Kebangsaan Malaysia, UKM, Bangi 43600, Selangor, Malaysia; shamsulk@ukm.edu.my; 6Analytical Services Laboratory, Chemistry and Technology Division, Malaysian Cocoa Board, Cocoa Innovation and Technology Centre, Lot 12621 Kawasan Perindustrian Nilai, Nilai 71800, Negeri Sembilan, Malaysia; badrul@koko.gov.my

**Keywords:** *Piper betle* L., *Piper rubro-venosum* hort. ex Rodigas, ^1^H-NMR, GC-MS, PCA, essential oils, acetic acid, chavicol acetate, chavibetol, chavibetol acetate

## Abstract

The differences in pungency of “sirih” imply the probable occurrence of several variants of *Piper betle* L. in Malaysia. However, the metabolite profiles underlying the pungency of the different variants remain a subject of further research. The differences in metabolite profiles of selected Malaysian *P. betle* variants were thus investigated; specifically, the leaf aqueous methanolic extracts and essential oils were analyzed via ^1^H-NMR and GC-MS metabolomics, respectively. Principal component analysis (PCA) of the ^1^H-NMR spectral data showed quantitative differences in the metabolite profiles of “sirih melayu” and “sirih india” and revealed an ambiguous group of samples with low acetic acid content, which was identified as *Piper rubro-venosum* hort. ex Rodigas based on DNA sequences of the internal transcribed spacer 2 (ITS2) region. The finding was supported by PCA of two GC-MS datasets of *P. betle* samples obtained from several states in Peninsular Malaysia, which displayed clustering of the samples into “sirih melayu” and “sirih india” groups. Higher abundance of chavicol acetate was consistently found to be characteristic of “sirih melayu”. The present research has provided preliminary evidence supporting the notion of occurrence of two *P. betle* variants in Malaysia based on chemical profiles, which may be related to the different genders of *P. betle*.

## 1. Introduction

In many Asian countries, chewing the leaves of betel vine (*Piper betle* L.) is a common social practice, largely embedded in the tradition, custom, or ritual of indigenous populations. The betel leaves are usually chewed together with areca nut and slaked lime, along with selected spices. The practice is believed to be beneficial in fighting bad breath (halitosis), while the leaves are also a traditional remedy for treating nosebleeds (epistaxis), and promoting pus removal [1,2]. Phylogenetic analyses have provided strong support for the placement of *P*. *betle* within the tropical Asian clade of *Piper*, specifically the west of Wallace’s line (WWL) clade. The WWL is a region that includes the Malay Peninsula, Sumatra, Java, and Borneo [3,4,5]. Locally known as “sirih” in Malay language, *P. betle* is an important plant in the Malaysian herbal industry [6,7]. Occurrence of several “sirih” variants, namely “sirih melayu”, “sirih cina”, and “sirih udang”, has been recorded in a transliterated manuscript of traditional Malay medicine [8].

Meanwhile, in India, there are more than 100 different variants of *P. betle*, which are termed landraces. A landrace is defined as a cultivated variety of plant, which has evolved in a geographical region as a result of its adaptation to the region’s edaphic and climatic conditions, and to the traditional management and consumption [9]. Since *P. betle* is a dioecious plant, the landrace is identified either as male or female based on its floral morphology [10]. A macroscopic evaluation of eight eastern Indian *P. betle* landraces emphasized that the variation between the landraces was apparent in terms of color and size of the leaves. The color of the leaves ranged from greenish yellow to dark green and the landraces with dark green leaves were larger in size compared to the landraces with greenish yellow leaves [11].

Previous mass spectrometry (MS) metabolomic studies of various Indian *P. betle* landraces have revealed clustering of leaf samples into several major groups based on the variation of metabolite profiles in intact plant materials, hexane extracts, and essential oils of dried and fresh leaves. Despite inconsistency in the reporting of one of the major metabolites that was identified either as eugenol or its isomer chavibetol, the studies reported several phenylpropenes, namely allylpyrocatechol, chavicol, and their acetate esters as the chemical markers of the resolved groups [10,12,13]. Phenylpropenes and their acetate esters are the characteristic constituents of *P. betle* leaf extracts that impart pungency upon chewing of the leaves [14].

While the difference between *P. betle* variants was physically noticeable from the color of the leaves, the variation was described by the individuals we communicated with as more pronounced in terms of their pungency. Since the notion of occurrence of different variants is solely based on human perception, a scientific investigation was deemed necessary to further investigate the association between varying pungency of the variants and their metabolite profiles. Hence, the present research employed a combination of ^1^H-NMR and GC-MS metabolomics to explore the variation in metabolite profiles of selected Malaysian *P. betle* variants and identify the chemical markers that may be utilized to discriminate between the different variants.

## 2. Results and Discussion

### 2.1. Principal Components Analysis of ^1^H-NMR Spectra of Three “Sirih” Variants

The investigation was initiated with a pilot experiment. Three “sirih” plants, growing in close vicinity to each other, were conveniently sampled from the herbal garden of Universiti Putra Malaysia (UPM) Agriculture Park. The plants were locally known as “sirih melayu” (*n* = 1), “sirih india” (*n* = 1), and “sirih merah” (*n* = 1) (Figure 1a). Six mature leaves were sampled from each plant. ^1^H-NMR spectra were acquired for the aqueous methanol-D4 (50% *v*/*v*) extracts of the leaf samples, followed by principal component analysis (PCA) of the spectral data. Representative ^1^H-NMR spectra of the samples are shown in Figure 1b and Appendix A. The intensity of the NMR signals at δ 1.92 and δ 2.08 for “sirih melayu” and “sirih india” was remarkably higher than those for “sirih merah”, and there were noticeable differences in the intensity of the signals in the sugar (δ 3.00–5.00) and aromatic (δ 6.00–8.00) regions for all three groups of samples. 

Prior to the PCA, the ^1^H-NMR spectra were converted to an 18 × 221 data matrix (18 samples of 221 variables) and the data matrix was preprocessed using three preprocessing methods commonly employed in metabolomics; i.e., unit variance (UV) scaling, Pareto scaling, and mean centering, to find the method that explained the highest percentage of variation in the first two principal components (PC) of the data matrix. Mean centering of the variables enabled the first two PCs to explain the highest percentage of variation in the data matrix (R^2^X [cumulative up to PC 2] = 94.9%), with predicted percentage of variation estimated by leave-one-out cross-validation (Q^2^X [cumulative up to PC 2]) of 87.8% (Appendix A). Mean centering the variables is advantageous because it removes an offset from the data matrix, reduces the model rank and improves the model fit [15]. Hence, the PCA model of mean-centered data matrix was selected for the interpretation of discriminating variables (Figure 1c).

Based on directions and magnitudes of the loading vectors in the first two PCs, PC scores of the samples were found to be heavily influenced by the δ 1.92, 3.76, and 3.80 variables (Appendix A). The δ 3.76 and 3.80 variables belong to methoxy protons. However, identification of the exact metabolites giving rise to the ^1^H-NMR signals binned into these variables was not trivial, as the methoxy protons may have belonged to any metabolite in the groups of *O*-methylated phenylpropanoids [16,17,18]. In contrast, the δ 1.92 variable, the intensity of which was remarkably high in “sirih india” and “sirih melayu”, was unequivocally identified as acetic acid (**1**), based on perfect automatic fit of the singlet at δ 1.92 to acetic acid in the Chenomx Profiler software library. The fitting also allowed quantification of acetic acid concentration in the samples relative to the concentration of internal standard 3-(trimethylsilyl)propionic-2,2,3,3-D4 acid sodium salt (TSP).

For the three groups of samples, at least one group had a statistically significantly different mean concentration of acetic acid (Welch’s *F* [2, 6.90] = 2805.50, *p* < 0.001, estimated ω^2^ = 1.00). A Games–Howell post hoc test indicated that the mean concentration (mM) of acetic acid in the leaves of “sirih merah” (mean = 0.36 [95% CI = 0.30, 0.43]) was significantly lower than that of the leaves of “sirih india” (mean = 10.94 [95% CI = 10.54, 11.34], mean difference = 10.57 [95% CI = −11.08, −10.07], *p* < 0.001) and “sirih melayu” (mean = 10.51 [95% CI = 9.88, 11.13], mean difference = 10.14 [95% CI = −10.93, −9.36], *p* < 0.001). However, the difference in the mean concentration of acetic acid in the leaves of “sirih india” and “sirih melayu” was not statistically significant (mean difference = 0.43 [95% CI = –0.38, 1.24], *p* = 0.339). Thus, relying only on the concentration of acetic acid to discriminate between the different variants of *P. betle* leaves was insufficient, because the mean concentration of acetic acid in “sirih india” was not statistically significantly higher than that of “sirih melayu”.

As shown in Figure 1c, acetic acid has strong positive correlation with variables in the aromatic region of the ^1^H-NMR spectra (δ 6.80 and 6.84) and variables that represent acetate protons of phenylpropene acetate esters (δ 2.04 and 2.08) [19]. In the biplot, these four variables are close to each other, thus providing support that they may belong to acetate esters of phenylpropenes, which are secondary metabolites commonly reported as constituents of *P. betle* leaf essential oil [20]. Meanwhile, the PC scores of “sirih merah” are heavily influenced by slightly different variables in the aromatic region (δ 6.28, 6.32, and 6.36, Appendix A), showing an abundance of a group of phenolic metabolites with different substituents on the benzene ring [21]. If the second PC of the biplot shows infraspecific variation within *P. betle*, then the first PC can be hypothesized to be an indication of a variation that is almost ten times greater than the infraspecific variation—infrageneric variation. While “sirih india” and “sirih melayu” were identified as *P. betle* based on morphology of the leaf and climbing habit of the plants, “sirih merah” was initially assumed to be a variant of *P. betle* because of similarities in climbing habit and shape of the leaves. Nevertheless, the findings from this pilot experiment revealed that “sirih merah” may be a different species of *Piper*, thus requiring further verification to confirm its identity.

Besides, further investigation of the variation of phenylpropenes and their acetate esters in the different *P. betle* variants requires more representative samples and specific analytical instruments of higher sensitivity and selectivity. For these reasons, the sample size was increased by sampling *P. betle* variants from selected locations in Peninsular Malaysia and the investigation was continued by GC-MS metabolomics of leaf essential oils of the different variants, thus allowing more focused analysis on the phenylpropenes and their acetate esters.

### 2.2. Identification of “Sirih Merah” as P. rubro-venosum hort. ex Rodigas

A literature search using the term “sirih merah” showed that the plant was identified either as *P. crocatum* Ruiz and Pav., *P. ornatum* N.E.Br., or *P. rubro-venosum* hort. ex Rodigas in Indonesia and Thailand [4,17,21,22,23,24,25,26,27]. Close examination of voucher specimens of the plant by our local botanist led to identification of the plant as *P.* cf. *fragile* Benth. Another botanist identified the plant as *P. fragile* and *P. ornatum* complex that displayed several leaf forms, and stated that the plant is often misidentified as *P. crocatum*, which is a shrubby species from Peru [28]. Other than that, previous research has reported color change of the leaves of the plant when it was planted in a new location [29]. Variations in the morphological characteristics have made the exact species identification of “sirih merah” difficult. In this respect, DNA barcoding, which is a method that takes advantage of the stability of DNA, could be a more effective method to differentiate and resolve closely related species [30]. Based on high similarity (99.39%) of the nuclear ribosomal DNA sequences of the internal transcribed spacer 2 (ITS2) region of “sirih merah” to a reference sequence in GenBank, the variant was identified as *P. rubro-venosum* hort. ex Rodigas (Figure 2 and Appendix A), which is an introduced species in Peninsular Malaysia and Singapore [31]. Despite being identified as a different *Piper* species, GC-MS data of *P. rubro-venosum* leaf essential oils were also included in GC-MS metabolomics to incorporate infrageneric variation into the PCA model. Only two samples of *P. rubro-venosum* leaf essential oils that were obtained from one plant (*n* = 1) were included as there was only one plant available at the time of sample collection and the number of leaves that fulfilled the selection criteria (healthy mature leaves of similar texture and color) allowed us to obtain at maximum only two samples of essential oils (see Appendix A).

### 2.3. Leaf Essential Oil Profiles of P. betle Variants and P. rubro-venosum

Further analysis was carried out using GC-MS to identify the leaf essential oil constituents of *P. betle* variants that were sampled in the present research, namely “sirih melayu”, “sirih india”, “sirih” unknown variant, and “sirih manis” (henceforth labeled as *P. betle* ‘melayu’, *P. betle* ‘india’, *P. betle* unknown variant, and *P. betle* ‘manis’) (Figure 3a) and *P. rubro-venosum* followed by PCA of the GC-MS data. In terms of data acquired for analysis of complex plant extracts, GC-MS provides three-dimensional (3D) data that include retention time and mass spectral information of metabolites detected in the extracts, whereas ^1^H-NMR provides 1D data, which is a combination of all ^1^H-NMR spectra of metabolites detected in the extracts. However, in electron ionization (EI) GC-MS, since the match scores, known as similarity index (SI) or reverse similarity index (RSI), against reference spectra in EI mass spectral libraries depend on (i) algorithms of the library search which vary between different manufacturers, and (ii) reproducibility of analysis, the correct metabolite does not necessarily occupy the first position in the list of hits [32,33]. Hence, relying only on high match scores may lead to misidentification of metabolites. 

Examination of both match scores and retention indices of other possible candidates in the list of hits is a powerful filtering method to reduce misidentification of metabolites in GC-MS [34]. Usually match scores of SI ≥80 and a linear retention index (LRI) that is within ±5 deviation from values reported in the literature are the guiding criteria for identification of metabolites. As another filtering method to reduce misidentification of metabolites, we also analyzed the leaf essential oil samples on two different GC-MS systems and checked for consistency of the metabolite profiles generated by the two systems. The basis for analyzing the samples using two GC-MS systems is that slightly different chromatograms may be acquired when a sample is analyzed using two different GC-MS systems, even though columns of similar dimension and polarity are used. This is due to the difference in a multitude of parameters of the GC-MS systems. For this reason, any generalization regarding metabolite profiles of the sample shall consider the metabolites that are consistently identified in both GC-MS systems.

The first GC-MS analysis (GC-MS analysis I) was carried out to examine whether the findings of the pilot experiment were also reflected in the leaf essential oil profiles of the *P. betle* ‘melayu’, ‘india’, and unknown variant, and *P. rubro-venosum* that were collected from six different locations in Peninsular Malaysia (Appendix A). Based on the results of GC-MS analysis I, the GC-MS parameters for the second analysis (GC-MS analysis II) were improved, which successfully detected a higher number of metabolites. The number of samples was increased by the inclusion of *P. betle* ‘manis’ in GC-MS analysis II. According to the locals, *P. betle* ’manis’ has the least pungency among the variants that were sampled in our research. The zoomed-in base peak chromatograms (BPC) of leaf essential oils of *P. betle* variants and *P. rubro-venosum* are provided in Appendix A and the identified essential oil constituents are listed in Appendix A [35,36,37,38,39,40,41,42,43,44,45,46,47,48].

Unlike the pungent aroma of *P. betle* leaf essential oils, the aroma of *P. rubro-venosum* leaf essential oils was notably non-pungent, with higher sweet notes. Representative BPCs for the essential oils of *P. rubro-venosum* leaves with green underside (RG) and maroon underside (RM), *P. betle* ‘melayu’ (Y1−Y5 and YA), *P. betle* ‘india’ (W1−W6, WA and WB), and *P. betle* unknown variant (XA) obtained in GC-MS analysis I are shown in Figure 3b. The BPCs show higher intensities of oxygenated monoterpenes linalool (**2**) and α-terpineol (**3**) peaks in the *P. rubro-venosum* leaf essential oil (RG) compared to the *P. betle* leaf essential oils. A high intensity of the chavicol acetate (**4**) peak was consistently measured only in *P. betle* ‘melayu’ in both GC-MS analyses, differentiating it from *P. betle* unknown variant, ‘india’, and ‘manis’. The BPCs also indicate that the latter three variants have similar leaf essential oil profiles, varying only in peak intensities. Chavibetol (**5**), chavibetol acetate (**7**), and 4-allyl-1,2-diacetoxybenzene (**8**) were found to be characteristic *O*-methylated phenylpropene and acetate esters of phenylpropenes in all *P. betle* leaf essential oils (Figure 3b,c).

Despite having similar sesquiterpene hydrocarbon germacrene D (**6**) peak intensities, the *P. rubro-venosum* leaf essential oils could be clearly distinguished from *P. betle* by the absence of chavibetol (**5**), chavibetol acetate (**7**), and 4-allyl-1,2-diacetoxybenzene (**8**) peaks. Of the 68 identified constituents, GC-MS analysis IIa of *P. rubro-venosum* leaf essential oils revealed the presence of only one acetate ester, namely (*E*)-chrysanthenyl acetate (Appendix A). Very low intensities of the signals at δ 2.08 and δ 6.80–6.92 in the ^1^H-NMR spectrum of *P. rubro-venosum* aqueous methanolic leaf extracts in the pilot experiment (Figure 1b) also reflected the lack/absence of the acetate esters of phenylpropenes in *P. rubro-venosum* leaves. 

In both GC-MS analyses, chavibetol (**5**) (LRI 1374.0 ± 1.4), instead of eugenol, was identified as one of the major constituents of *P. betle* leaf essential oils. Eugenol (LRI 1361.7 ± 0.5) and (*Z*)-isoeugenol (1391.8 ± 0.5), which were detected only in GC-MS analysis II, were identified as minor constituents. Another derivative of eugenol, methyl eugenol (LRI 1410.3 ± 0.5), was detected as a minor constituent in both GC-MS analyses (Appendix A). Initially, for the peak with LRI 1374.0 ± 1.4, eugenol, 3-allyl-2-methoxyphenol, (*Z*)-2-methoxy-4-(prop-1-en-1-yl)phenol, and (*Z*)-2-methoxy-4-(prop-1-en-1-yl)-phenol were the top candidates in the list of hits with match scores ranging from 90 to 95. However, the LRIs of these constituents were found to deviate by more than ±5 when compared to the retention indices reported in the main references [35,36]. Moreover, chavibetol or 5-allyl-2-methoxyphenol was not among the top candidates of the hit list. Further examination of other top candidates in MassHunter’s list of hits located chavibetol, with a match score of 85 and LRI that matched literature data [46]. In the literature, either eugenol or its isomer chavibetol were reported as major constituents and chemical markers of *P. betle* leaf extracts [19,20,47,48,49,50,51,52,53,54]. Apparently, some studies that reported eugenol as the major constituent of *P. betle* leaf extracts based the identification solely on high match score or comparison with literature data without application of any other filtering method [12,19,49,54].

In the *Malaysian Herbal Monograph*, eugenol is listed as the chemical marker of *P. betle* leaf, and an HPLC method has been developed to confirm its presence in leaf extracts of the herb [7]. However, eugenol and its derivatives are also the chemical markers of several other spices, such as dried barks of *Cinnamomum verum* and *C. cassia* (Lauraceae), dried flower buds of *Syzygium aromaticum* (Myrtaceae), and leaves of *Ocimum sanctum* (Lamiaceae) [55,56,57], to which *P. betle* is distantly genetically related. This raises some doubts on the suitability of using eugenol as the chemical marker of *P. betle* leaf. Moreover, previous research that used both HPLC and GC-MS has shown that the difference in retention times between the eugenol and chavibetol standard is more apparent when analyzed using GC-MS (0.3 versus 0.6 min), and the retention time of the major peak in the GC-MS chromatogram of *P. betle* leaf extract also showed a better match with that of chavibetol standard, rather than eugenol, which appeared as a small peak preceding that of chavibetol [58]. Similarly, in our research, eugenol was shown to be a minor constituent of *P. betle* leaf essential oils, appearing as a small peak (peak number 15 in Appendix A) immediately before the much higher intensity peak of chavibetol in the GC-MS chromatogram, thus supporting our finding that chavibetol is the major constituent of *P. betle* leaf essential oils, while eugenol is a minor constituent. The same situation applies for the identification of chavibetol acetate (**7**) (LRI 1532.5 ± 0.7), which was identified as another major constituent instead of eugenol acetate (see Appendix A). Based on our findings, we propose that chavibetol (**5**) and chavibetol acetate (**7**) are more suitable for use in pharmacognostical analysis of *P. betle* leaves. 

### 2.4. Principal Components Analysis of GC-MS Data of P. betle Leaf Essential Oils

Despite our efforts to identify as many constituents as possible using two different GC-MS systems, several small peaks could not be detected or identified by the integrators in the GC-MS data processing software. Hence, for PCA of the GC-MS data, all BPCs from both GC-MS analyses were matricized to avoid any exclusion of undetected or unidentified peaks. This method is different from those used in previous studies, which performed PCA only on peak areas or peak intensities of constituents that were identified in *P. betle* samples [10,12,59]. Figure 4a shows that the *P. betle* of unknown variant (sample XA) is clustered together with *P. betle* ‘india’ (sample W1−W6, WA, and WB) in the PCA biplot of GC-MS analysis I data. The first PC shows that chavibetol (**5**), chavibetol acetate (**7**), and 4-allyl-1,2-diacetoxybenzene (**8**) are the discriminant factors for *P. betle* leaf essential oils, whereas α-terpineol (**3**) and terpinen-4-ol are the discriminant factors for *P. rubro-venosum* leaf essential oils. Linalool (**2**) was not identified as a discriminant factor for *P. rubro-venosum* as the intensity of the linalool peak was found to be high only in RG and not RM.

The second PC of GC-MS analysis I data shows that the higher intensity of methyl salicylate is characteristic of *P. betle* ‘india’ and higher intensity of chavicol acetate (**4**) and methyl chavicol peak are characteristic of *P. betle* ‘melayu’ (samples Y1−Y5 and YA). However, in the PCA of GC-MS analysis II data (Figure 4b) only chavicol acetate (LRI 1347.0 ± 1.4) was identified as the discriminant factor for *P. betle* ‘melayu’. In GC-MS analysis I, chavicol acetate was not detected in *P. betle* ‘india’ and unknown variant, but in GC-MS analysis II chavicol acetate was detected in low intensities in both variants and *P. betle* ‘manis’. These differences in detection capability between the analyses could be attributed to the use of split and splitless injections in GC-MS analyses I and II, respectively, and higher sensitivity of the GC-MS system used in GC-MS analysis II. In the first PC of GC-MS analysis II data, the essential oil sample of *P. betle* ‘melayu’ leaves collected from Kepong, Selangor (sample YA) clustered together with the essential oil samples of *P. betle* ‘melayu’ leaves collected from Kuantan, Pahang (sample Y1−Y5), indicating the high loading of chavicol acetate in the first PC scores of all *P. betle* ‘melayu’ samples, regardless of their geographical origins. Consequently, other *P. betle* samples with low loadings of chavicol acetate clustered together, as in the case of *P. betle* ‘india’, unknown variant, and ‘manis’. On the other hand, the second PC of GC-MS analysis II data could be related to the variation in intensities of the constituent peaks in the BPCs as illustrated in Figure 3c, where it can be observed that the intensities of constituent peaks in *P. betle* ‘india’ are generally higher than those in *P. betle* ‘manis’. Considering the distribution of the second PC scores of all *P. betle* ‘melayu’ samples in the biplot of GC-MS analysis II data, it is reasonable to group the other samples (W1−W6, WA, WB, XA, and M1−M6) as *P. betle* ‘india’.

### 2.5. Hypotheses

The PCA of ^1^H-NMR spectra and GC-MS data revealed a paucity of acetic acid and acetate esters of phenylpropenes in *P. rubro-venosum* leaves, respectively. Other than *P. rubro-venosum* leaves, analysis of ^1^H-NMR spectra of *P. sarmentosum* aqueous methanolic leaf extracts, which were morphologically similar to *P. betle* leaves, also showed the paucity of acetic acid and acetate esters of phenylpropenes [60]. In contrast, the abundance of acetic acid in aqueous methanolic leaf extracts and *O*-methylated phenylpropene chavibetol and acetate esters of phenylpropenes, namely chavicol acetate, chavibetol acetate, and 4-allyl-1,2-diacetoxybenze, in the essential oils were found to be characteristic of *P. betle* leaves, regardless of the variants. Leaf essential oils of several species in the genus *Pimenta* (Myrtaceae), of which some are used as spices, were reported to contain chavibetol as major constituents. However, identification of acetate esters of phenylpropenes in the leaf essential oils of these species has never been reported in the literature [37,61,62,63]. Acetic acid is a precursor of acetyl-CoA, which is a metabolic intermediate responsible for the biosynthesis of acetylated metabolites and isoprenoids in plants [64,65]. Hence, we hypothesized that biosynthesis of acetate esters of phenylpropenes is associated with pooling of acetic acid, which is a part of specialized metabolism in *P. betle* leaves. 

In general, consumption of *P. betle* ‘india’ is preferred over ‘melayu’ due to its less pungent flavor, which may be partly attributed to the remarkably lower level of chavicol acetate compared to *P. betle* ‘melayu’. Identification of chavicol acetate as a chemical marker of a cluster of *P. betle* samples in PC scores space is parallel with the findings of previous research, which showed clustering of several Indian *P. betle* landraces into two major groups based on the intensity of the chavicol acetate peak [59]. Other than that, analyses using DNA markers of *P. betle* samples have revealed two major clusters based on gender [66,67]. Macroscopic examination for gender distinction of *P. betle* can be made only after flowering, based on morphology of the inflorescences (termed spikes) [10,66]. Male spikes are generally longer than female spikes [66]. As exemplified in our sampling, gender distinction of *P. betle* remains a challenging task because only *P. betle* ‘melayu’ collected from Kuantan, Pahang, and *P. betle* ‘india’ collected from Raub, Pahang bore the spikes (Appendix A). The spikes of *P. betle* ‘melayu’ and ‘india’ approximately matched the description of morphological characteristics of male and female spikes, respectively [66]. Both variants were resolved into two major groups in the first PC due to higher abundance of chavicol acetate in *P. betle* ‘melayu’ leaf essential oils. For these reasons, it was hypothesized that *P. betle* ‘melayu’ is the male plant whereas *P. betle* ‘india’ is the female plant, and chavicol acetate may be used as a chemical marker to differentiate between the two Malaysian *P. betle* variants and the two genders of this dioecious perianthless Piperales plant.

## 3. Materials and Methods

### 3.1. Chemicals and Disposables

Methanol-D4 (99.8% minimum deuteration degree), deuterium oxide (99.9% minimum deuteration degree) and 3-(trimethylsilyl)propionic-2,2,3,3-D4 acid sodium salt (98% minimum deuteration degree) purchased from Merck KGaA (Darmstadt, Germany) were used in the preparation of extracts in ^1^H-NMR metabolomics. Liquid chromatography grade *n*-hexane (≥98.0% purity) and *n*-nonane (99.3 ± 0.1% purity) were purchased from Fisher Scientific, Leicestershire, United Kingdom and CPAchem, Stara Zagora, Bulgaria, respectively. Alkanes standard (C7–C33) was purchased from Restek Corporation, Bellefonte, PA, USA. PTFE syringe filters (13 mm, 0.22 µm) were purchased from Orioner Hightech, Kuala Lumpur, Malaysia.

### 3.2. ^1^H-NMR Metabolomics of “Sirih” Leaf Extracts

#### 3.2.1. Collection of Plant Materials

In the present research, the term ‘variant’ is used instead of landraces to indicate that the names are not based on the specific geographical region where the plants were grown. Firstly, fresh leaves with intact petioles of three local variants of “sirih”, known as “sirih merah”, “sirih india”, and “sirih melayu”, were collected on 5 July 2018 from the herbal garden within the University Agriculture Park, Universiti Putra Malaysia, Serdang, Selangor, Malaysia. Coordinates of sampling locations are listed in Table 1. For each variant, a total of six leaves were collected from a plant to represent six replicates. The six leaves collected were similar in size and color and grew at the 11th or 12th node of different stems [12]. Collection of the leaves was carried out between 10:15 and 10:55 a.m. at a temperature of 35 °C and relative air humidity of 76%. Voucher specimens were deposited in the herbarium of Universiti Kebangsaan Malaysia (UKMB).

#### 3.2.2. Preparation of Samples and Extracts

Methods of sample preparation and extraction were adapted from the protocol of plant NMR metabolomics [68]. Immediately after harvesting, the leaves were wiped clean with dry tissue paper and the petioles were removed manually. The leaf blades (without midvein) were then quenched in liquid nitrogen and ground into powder form using a mortar and pestle. The mortar and pestle were wiped clean between samples to minimize cross contamination. The powdered samples were then transferred into 2 mL centrifuge tubes, frozen at −80 °C for 24 h and lyophilized for 24 h. The lyophilized samples were kept in a desiccator, away from light, prior to extraction.

Thirty (30) mg of lyophilized sample was extracted with 1.2 mL deuterated solvent made up of 600 µL methanol-D4 and 600 µL phosphate buffer (90 mM, pH 6.0) in deuterium oxide containing 0.1% TSP as internal standard, using a vortex mixer for 1 min. Metabolites in the sample were further extracted by sonication for 15 min at 35 kHz and room temperature in an ultrasonicator (LC 60/H, Elma Schmidbauer GmbH, Singen, Germany) followed by centrifugation at 13,000 rpm for 15 min (Hettich^®^ MIKRO 20, Andreas Hettich GmbH & Co. KG, Tuttlingen, Germany). The supernatant was filtered through a 0.22 µm PTFE syringe filter and 600 µL was transferred into 5 mm NMR tubes [69].

#### 3.2.3. Acquisition of ^1^H-NMR Spectra

Firstly, all extracts were left at room temperature for 30 min. Next, NMR spectral data were acquired at 24 °C on an Agilent VNMRS500 spectrometer system equipped with a narrow bore premium shielded magnet operating at a frequency of 500 MHz, with a 5 mm inverse detection pulsed field gradient (ID/PFG) probe, and VnmrJ 6.1C software (Agilent Technologies, Santa Clara, CA, USA). Acquisition parameters are described herein: solvent lock = deuterium oxide, spectral width = −2.00–14.00 ppm, number of scans = 8, relaxation delay = 1 s, pulse angle = 45° and receiver gain = 32 dB. Presaturation pulse sequence (PRESAT) was utilized to suppress the residual water signal with the following parameters: number of scans = 64 and presaturation delay = 2 s. The FID was transformed to NMR spectrum using Fourier transform with weighting function to enhance the resolution (command wft), followed by automatic phasing of both zero- and first-order terms (command aph). A total of 18 extracts were analyzed.

#### 3.2.4. Preprocessing of ^1^H-NMR Spectra 

NMR spectra of all samples were superimposed to identify meaningful regions (δ 0.50–9.70) and residual solvent signals (methanol: δ 3.27–3.33 and water: δ 4.70–5.07) using MestReNova version 6.0.2-5475 (Mestrelab Research S.L., Santiago de Compostela, Spain). The NMR spectra were further processed using Chenomx Processor version 8.2 (Chenomx, Edmonton, AB, Canada) with the following parameters: chemical shape indicator (CSI) = TSP, TSP concentration = 0.2902 mM, sample pH = 6.0 ± 0.5, and automatic phase, baseline (cubic spline), and shim correction. Identification and quantification of metabolites were carried out by fitting the NMR spectra to reference spectra in Chenomx Spectral Reference Library in Chenomx Profiler version 8.2. Identity and estimated concentration of a metabolite was considered valid only in case of all NMR signals of the metabolite fitting perfectly to all signals of the metabolite in the reference library and no overlapping with signals of other metabolites. The NMR spectra were normalized to total area and the residual solvent signals were removed. Next, the meaningful regions were discretized to buckets of δ 0.04 width, producing a data matrix dimension of 18 × 221 in tsv file format.

### 3.3. DNA Barcoding of “Sirih Merah”

The sequence of ITS2 region was selected for the DNA barcoding analysis due to its length, which is shorter than that of full-length ITS (average length = 233 versus 634 base pairs), thus less prone to amplification and sequencing errors for rapid identification of plant species [70]. DNA barcoding was performed following the protocol described in previous research [30]. Briefly, whole cell DNA was extracted twice using the conventional cetyltrimethylammonium bromide (CTAB) procedure and purified using a High Pure PCR template preparation kit (Roche Molecular Systems, Inc., Pleasanton, CA, USA). DNA barcodes were generated using the primer pairs ITS2-2F and ITS2-3R [71] and amplified in 10 μL reaction mixture, comprising 0.3 μM of each primer, 10 ng of template DNA, and one Type-it Multiplex PCR master mix (Qiagen, Germantown, MD, USA) using GeneAmp PCR System 9700 (Thermo Fisher Scientific, Waltham, MA, USA). Next, the PCR products were bidirectionally sequenced using a BigDye Terminator v3.1 cycle sequencing kit (Thermo Fisher Scientific, Waltham, MA, USA) based on the Sanger chain-termination method, purified using a BigDye XTerminator purification kit (Thermo Fisher Scientific, Waltham, MA, USA) and analyzed using ABI3130xl Genetic Analyzer (Thermo Fisher Scientific, Waltham, MA, USA). ITS2 sequence data was edited using Sequencher version 5.0 (Gene Codes Corporation, Ann Arbor, MI, USA) and matched against reference sequences in the NCBI GenBank database using Basic Local Alignment Search Tool (BLAST) and a neighbor-joining (NJ) tree. The NJ tree was constructed based on Kimura two-parameter (K2P) divergences and 1000 bootstrap replications in MEGA6 software [72].

### 3.4. GC-MS Metabolomics of Leaf Essential Oils

#### 3.4.1. Collection of Plant Materials

Our efforts to collect samples of the three *P. betle* variants described in the transliterated manuscript of traditional Malay medicine [8] were only partly successful. From the three variants, we managed to obtain only *P. betle* ‘melayu’ after communications with several Malaysian locals and *P. betle* farmers. Apart from *P. betle* ‘melayu’, we collected samples locally known as *P. betle* ‘india’, ‘manis’, and an unknown variant that morphologically resembled *P. betle* ‘india’. Mature leaves of similar texture and color with intact petioles of *P. betle* ‘melayu’ (sample code Y1−Y5 and YA), ‘india’ (W1−W5, WA and WB), ‘manis’ (M1−M6), unknown variant (XA), and *P. rubro-venosum* (RM and RG) were collected from six different locations, comprising three states in Peninsular Malaysia (Appendix A). Details on collection date and time, sampling locations, growing conditions, and occurrence of inflorescence are listed in Appendix A. Following the local practice of maintaining the freshness of harvested *P. betle* leaves, immediately after harvesting, the leaves were transported to the laboratory. The leaves were stored in plastic bags at room temperature, lightly sprayed with tap water daily, and processed within five days after harvesting. 

#### 3.4.2. Hydrodistillation of Plant Materials

Prior to hydrodistillation, the leaves were examined for undesired physical attributes, which might be inflicted upon collection, transportation, or storage. Leaves of unsatisfactory quality were excluded (Appendix A). Next, the petioles of the leaves were removed, and the leaf blades were cut in slices using clean scissors. Depending on the number of good quality leaves, varying amounts of sliced leaf blades were loaded into a 10 L round-bottomed flask filled with 7 L tap water (Appendix A). Hydrodistillation was carried out for 2 h, where the starting time (t_0_) was the moment the heater of the apparatus being switched on [73]. Two samples were hydrodistilled per day, with thorough washing of the Clevenger-type apparatus between the samples. Subsequently, 10 mL HPLC grade *n*-hexane was added to the distillate in a separatory funnel. The *n*-hexane layer was collected after 18 h of partitioning, followed by evaporation of the *n*-hexane in a fume hood. Next, the essential oils obtained were weighed and stored in tightly sealed containers at 4 °C until further analysis.

#### 3.4.3. GC-MS Analyses

GC-MS analyses were performed using two gas chromatography systems manufactured by different manufacturers, two columns (dimethylsilicone stationary phase with 5% phenyl groups) by two manufacturers, and two sets of electron ionization mass spectral libraries. The analytical parameters for analysis I were an improvisation of standard procedure for analysis of essential oils in our laboratory while the parameters for analysis II were optimized from the findings of GC-MS analysis I.

##### GC-MS Analysis I

The gas chromatography system used for GC-MS I was GCMS-QP2010 Ultra (Shimadzu, Kyoto, Japan). The mass spectrometer was operated in electron ionization mode at 70 eV, ion source temperature = 200 °C, scanning mode = full scan, scan rate = 10 scans/s, mass range = 40–700 Da, and solvent delay = 2.55 min. The inlet temperature was set at 250 °C. Firstly, 10 µL of the essential oil was diluted in 2 mL LC grade *n*-hexane. One (1) µL of the diluted essential oil was injected in split mode (10:1) into a Rxi-5ms column of 30.00 m length × 0.25 mm internal diameter × 0.25 µm film thickness (Restek Corporation, Bellefonte, PA, USA). Helium flow rate was constant at 0.80 mL/min.

The oven temperature was programmed as the following: 50 °C (1.0 min), 4 °C/min to 130 °C (0.0 min), 1 °C/min to 170 °C (0.0 min), 5 °C/min to 240 °C (5.0 min), making a total run time of 80 min. The order of run sequence was not randomized, and solvent blank (*n*-hexane) was injected at the beginning of the run sequence and after every injection of sample to ensure no carryover from the previous sample. Alkanes standard was injected after the first solvent blank injection for calculation of linear retention indices (LRI) using the H. Van den Dool and P. Dec Kratz equation [74].

For the identification of essential oil constituents, EI mass spectra were matched against reference spectra in Wiley Registry 9, National Institute of Standards and Technology (NIST) 2011 and Wiley’s Flavors and Fragrances of Natural and Synthetic Compounds (FFNSC) 1.3 EI mass spectral libraries, and the threshold of similarity index (SI) between measured and reference spectra was set at 80. A hit list of top 10 best candidates for each constituent was generated in pdf file format using GCMSsolution 2.72 software (Shimadzu, Kyoto, Japan). For both GC-MS analyses, the difference between calculated LRI and literature data was considered to minimize false positives. Hence, only constituents with a LRI within ±5 deviation from the LRI of literature data were reported [75]. The main references for the identification of essential oil constituents were compilations of LRI of frequently reported essential oil constituents [35,36] and only in case where the LRI did not match the LRI reported in the main references were other studies referred to [37,38,39,40,41,42,43,44,45,46,47,48].

##### GC-MS Analysis II

The gas chromatograph used for GC-MS II was the 7890A Gas Chromatograph equipped with 7000 Series Triple Quadrupole Mass Spectrometer (Agilent Technologies, Santa Clara, CA, USA). The mass spectrometer was operated in EI mode at 70 eV and at ion source temperature of 200 °C. The following parameters were set for data acquisition: mode = full scan, scan rate = 20 scans/s, mass range = 40–250 Da and solvent delay = 5 min. The inlet temperature was set at 250 °C. An aliquot of 2.5 µL essential oil was diluted in 1 mL LC grade *n*-hexane containing 5 ppm *n*-nonane as internal standard. A pooled quality control (QC) sample was prepared by mixing equal aliquots of each sample. One (1) µL of the diluted essential oil was injected in splitless mode into a HP-5MS column of 30.00 m length × 0.25 mm internal diameter × 0.25 µm film thickness (Agilent Technologies, Santa Clara, CA, USA). Helium flow rate was maintained at 0.80 mL/min. 

The oven temperature was programmed as the following: 50 °C (1.0 min), 4 °C/min to 126 °C (0.0 min), 1 °C/min to 148 °C (0.0 min), 100 °C/min to 250 °C (0.0 min), making a total run time of 43.02 min. Post-run column temperature was set at 310 °C for a period of 1 min, to elute out remaining sample prior to injection of the next sample. A solvent blank (*n*-hexane containing 5 ppm *n*-nonane) was injected at the beginning of the run sequence. The alkanes standard was injected after the first solvent blank injection and the end of the run sequence for calculation of LRI and examination of retention time shift [76]. The order of run sequence was randomized manually by drawing cards of sample codes from a bowl. After injection of four to six samples, one QC sample was injected throughout the run sequence.

Identification of essential oil constituents was carried out using MassHunter Workstation Qualitative Analysis Workflows B.08.00 software (Agilent Technologies, Santa Clara, CA, USA). The following parameters were set: identification algorithm = peak integration, integrator = Agile 2 and minimum integrated area = 0.05% of largest peak in the total ion current (TIC) chromatogram. Mass spectra of constituents were matched against 15 best candidates in NIST 2014 and Wiley’s FFNSC 3.0 EI mass spectral libraries with minimum reverse similarity index (RSI) threshold set at 80. Only constituents with a LRI within ±5 deviation from the LRI of literature data were reported [75].

#### 3.4.4. Preprocessing of GC-MS Data

Raw data in Shimadzu QGD (GC-MS analysis I) and Agilent D (GC-MS analysis II) file format were converted to netCDF using GCMSsolution 2.72 (Shimadzu, Kyoto, Japan) and GC/MS Translator (Agilent Technologies, Santa Clara, CA, USA), respectively. The netCDF files were exported to MZmine version 2.53 software for visualization of chromatograms [77]. Two- and three-dimensional base peak chromatograms (BPC) were visualized to aid the input of parameters in the MS-DIAL version 4.24 software [78].

Trial and error was used to optimize the parameters in the MS-DIAL. The optimized parameters for peak detection, deconvolution, alignment, and filtering, including parameters selected for results export, are listed in Appendix A. Aligned peak lists for both GC-MS analyses were normalized to TIC, producing data matrices in txt file format. Subsequently, the data matrices were exported to Microsoft Office Excel 2016. Next, preparation of the data matrices (exclusion of variables, transposing, and labeling) for principal component analysis (PCA) was carried out using Microsoft Office Excel 2016. All variables in the data matrix of GC-MS I were retained while only variables with less than 30% relative standard deviation (<30% RSD) in all four QC samples of GC-MS II were retained [79].

### 3.5. Welch’s Analysis of Variance (ANOVA) and Principal Components Analysis (PCA)

Welch’s ANOVA with Games–Howell post hoc test was performed using SPSS version 26 software (International Business Machines Corporation, New York, NY, USA). PCA was carried out using SIMCA-P 14.1 software (Sartorius Stedim Data Analytics AB, Umeå, Sweden). Firstly, normal distribution of variables was assessed based on skewness and ratio of minimum–maximum values of the variables. In case of skewed distribution of most variables in a data matrix (as in the GC-MS data matrices), the whole data matrix was log-transformed [80]. Variables that could not be log-transformed were treated as missing data, thus giving no influence on the final PCA model. Unit variance (UV) and Pareto scaling and mean centering were performed on the data matrices to find an operation that allowed the capture of the highest percentage of variation explained by the first two principal components (highest R^2^X [cumulative up to PC 2]) [81]. Leave-one-out cross validation procedure was used for the ^1^H-NMR and GC-MS analysis I data matrices whereas SIMCA-P’s default seven-groups cross validation procedure was used for the GC-MS analysis II data matrix [82,83]. To facilitate interpretation of PCA loadings, only well-modeled variables were made visible in loading plots and biplots. Well-modeled variables were defined as the variables that were well-predicted by the first two principal components, as estimated by cross-validation (Q^2^VX [cumulative up to PC 2] ≥80%).

## 4. Conclusions

Utilization of 1D ^1^H-NMR metabolomics at the initial stage of the present research resolved the ambiguity in identification of Malaysian *P. betle* variants, where the significantly lower concentration of acetic acid and intensity of ^1^H-NMR signals for acetate esters of phenylpropenes in a group of samples necessitated its verification using DNA barcoding of ITS2 sequences. The group was identified as *P. rubro-venosum*, which originates from Indonesia and is considered as an introduced species in Malaysia. Results from the principal component analysis of the ^1^H-NMR spectral data directed further detailed analysis using GC-MS metabolomics. Using two 3D datasets, GC-MS metabolomics confirmed the absence of acetate esters of phenylpropenes in *P. rubro-venosum* leaf essential oils and revealed the occurrence of two *P. betle* variants, namely *P. betle* ‘melayu’ and ‘india’, which could be distinguished based on the abundance of chavicol acetate in their leaf essential oils. The abundance of acetic acid and acetate esters of phenylpropenes in the leaves were hypothesized to be a part of a specialized metabolism in *P. betle*, and that *P. betle* ‘melayu’ and ‘india’ are the male and female *P. betle*, respectively. Nevertheless, more detailed studies involving larger sample sizes and wider coverage of geographical distribution, as well as controlled experimentation, shall be carried out to test the hypotheses and provide better insights into the essential oil biochemistry of this medicinal plant.

## Figures and Tables

**Figure 1 plants-10-02510-f001:**
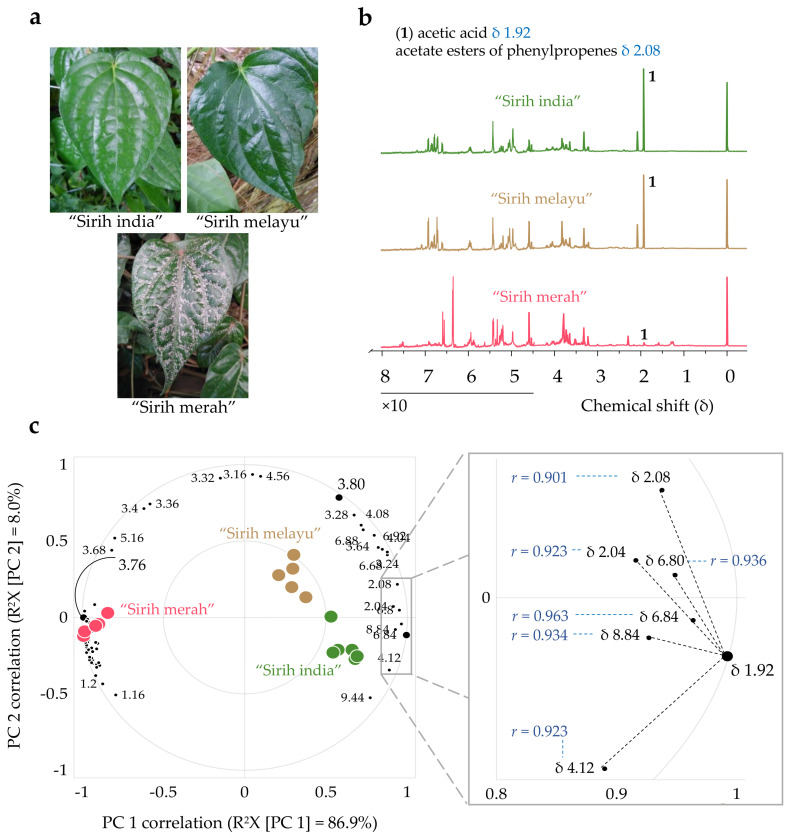
Principal component analysis (PCA) of ^1^H−NMR spectra of “sirih” leaf aqueous methanolic extracts. (**a**) Leaf of “sirih india”, “sirih melayu”, and “sirih merah” sampled in the pilot experiment. (**b**) Representative ^1^H−NMR spectra of the three “sirih” variants showing that acetic acid (**1**) and acetate esters of phenylpropenes are the major metabolites in the leaf aqueous methanolic extracts of “sirih india” and “sirih melayu” but remarkably low in “sirih merah”. The spectra were normalized to TSP peak at δ 0.00 (intensity of TSP peak = 100). The intensity at δ 4.50−8.00 was increased tenfold to help visualize the signals in the olefinic and aromatic regions. The full spectra are shown in Appendix A. (**c**) Correlation−scaled PCA biplot of mean−centered data matrix showing only well−modeled variables (cumulative percentage of variation of the variable predicted by the first two PCs, as estimated by cross−validation (Q^2^VX [cumulative up to PC 2]) ≥80%). The zoomed−in region indicates the variables that have strong positive correlation with the δ 1.92 variable (*r* > 0.900). Complete labeling of characteristic variables of “sirih merah” can be viewed in Appendix A.

**Figure 2 plants-10-02510-f002:**
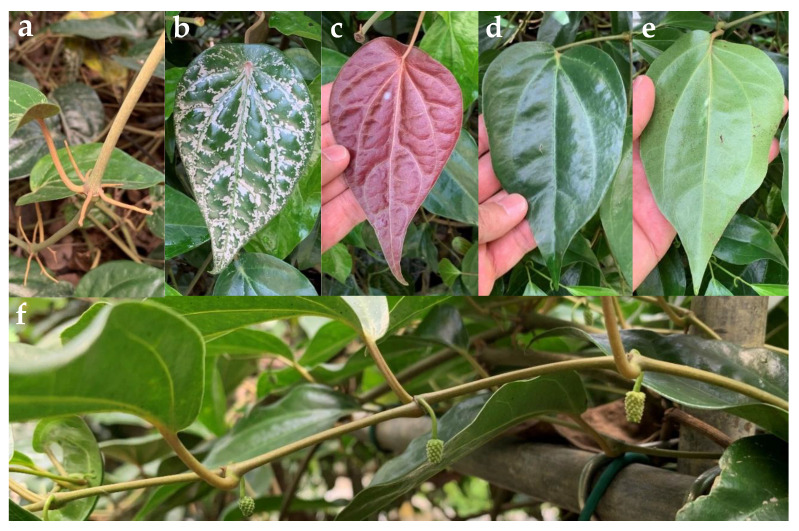
Photographs of *Piper rubro*−*venosum* hort. ex Rodigas. (**a**) Nodal roots; (**b**) adaxial surface of silver pink leaf, showing silver pink stripes mostly along primary veins; (**c**) maroon abaxial surface of leaf (**b**), (**d**) adaxial surface of plain green leaf; (**e**) green abaxial surface of leaf (**d**); (**f**) leaf-opposed inflorescences. (**b**,**c**) are designated as RM whereas (**d**,**e**) are designated as RG.

**Figure 3 plants-10-02510-f003:**
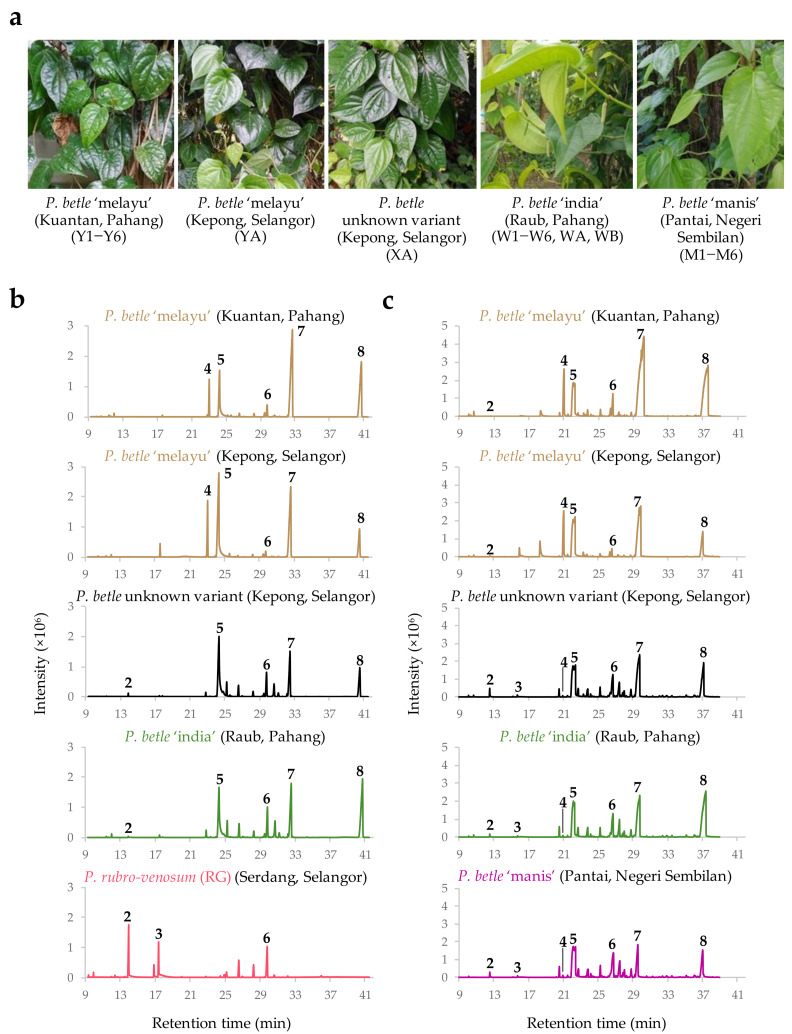
GC−MS analysis of *Piper betle* and *Piper rubro*−*venosum* leaf essential oils. (**a**) Photographs of *P. betle* ‘melayu’, unknown variant, ‘india’, and ‘manis’. (**b**) Representative base peak chromatograms (BPC) of leaf essential oils in GC−MS analysis I and (**c**) GC−MS analysis II; (**2**) linalool, (**3**) α−terpineol, (**4**) chavicol acetate, (**5**) chavibetol, (**6**) germacrene D, (**7**) chavibetol acetate, (**8**) 4−allyl−1,2−diacetoxybenzene.

**Figure 4 plants-10-02510-f004:**
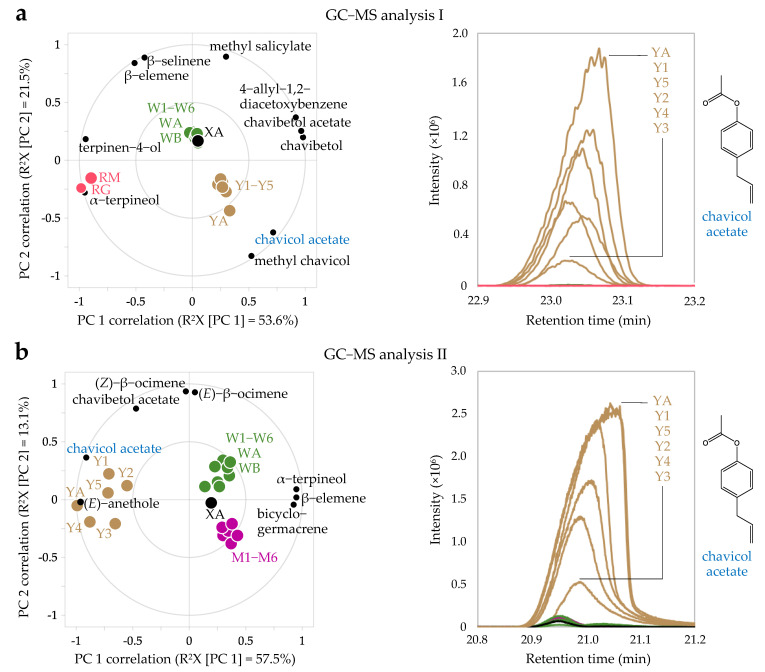
Principal component analysis (PCA) of matricized, logarithm (base 10)−transformed, and mean−centered base peak chromatograms (BPC) of *Piper betle* and *Piper rubro*−*venosum* leaf essential oils. *P. rubro*−*venosum* leaves with green underside (RG) and maroon underside (RM), *P. betle* ‘melayu’ (Y1−Y5 and YA), *P. betle* ‘india’ (W1−W6, WA, and WB), *P. betle* unknown variant (XA), and *P. betle* ‘manis’ (M1−M6). Only well−modeled variables are shown (Q^2^VX [cumulative up to PC 2] ≥80%). (**a**) Correlation−scaled PCA biplot of GC−MS analysis I data (Q^2^X [cumulative up to PC 2] = 61.8%) and chavicol acetate peak in the BPC of GC−MS analysis I. (**b**) Correlation−scaled PCA biplot of GC−MS analysis II data (Q^2^X [cumulative up to PC 2] = 63.4%) and chavicol acetate peak in the BPC of GC−MS analysis II. Considerations for selection of PCA models can be viewed in Appendix A.

**Table 1 plants-10-02510-t001:** Coordinates of sampling locations of the “sirih” variants.

Sample	Voucher Specimen No.	Coordinates	Altitude (m)
“sirih merah”	UKMB404018	2°59′21.4″ N, 101°42′30.3″ E	50
“sirih india”	UKMB404019	2°59′19.4″ N, 101°42′30.0″ E	51
“sirih melayu”	UKMB404020	2°59′19.6″ N, 101°42′29.7″ E	49

## Data Availability

Metabolomics data have been deposited in the EMBL-EBI MetaboLights database with identifier number MTBLS2585. ITS2 sequence of “sirih merah” was deposited in the NCBI GenBank database with accession number OK422240.

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
