# Peer review of "Metabolomics-Driven Discovery of an Introduced Species and Two Malaysian Piper betle L. Variants"

_plants, 2021, doi:10.3390/plants10112510_

Round 1

Reviewer 1 Report

Good job

  • You have collected mature leaves. It would be better if you had leaf samples from each age class (young, mature, and old) and more samples.
  • Several studies found that plant species growing under different environmental conditions (e.g., altitude and soil nutrients) show significant differences in the composition of the primary and secondary metabolite composition. In table 1 you reported the altitude; if it is possible, please report the soil conditions and agricultural practices.
  • In Table S1. In column “Growing condition” instead of “Grown under shade” should better had PAR measurements (photosynthetically active radiation). 

Author Response

Point 1: Good job. You have collected mature leaves. It would be better if you had leaf samples from each age class (young, mature, and old) and more samples.

Response 1: Thank you for the suggestion and we will consider sampling from different age classes in future research. In the present research, we are interested to find out if the notion of occurrence of different variants of Malaysian P. betle consumed by local community can be illustrated in the principal components. From our communication with P. betle farmers, only mature leaves were harvested for sale at local markets. Hence, we fixed the age of samples by sampling only mature leaves, which are the ones that are harvested by the farmers and consumed by the local community.

We acknowledge the advantage of carrying out research with large sample size in terms of higher statistical power, thus more conclusive results. Nonetheless, in our case, obtaining essential oils from fresh leaves of different age class means we will have more plant materials to be hydrodistilled. This endeavour is hardly practical because there is higher probability of introducing undesired variation due to longer storage period for some of the fresh leaves prior to hydrodistillation. As stated in the Method section, our hydrodistillation protocol that was designed to minimize loss of more volatile essential oil constituents (such as the monoterpenes) and reduce cross contamination between samples limited our capability to hydrodistillation of two samples per day (page 13, lines 569–570).

Point 2: Several studies found that plant species growing under different environmental conditions (e.g., altitude and soil nutrients) show significant differences in the composition of the primary and secondary metabolite composition. In Table 1 you reported the altitude; if it is possible, please report the soil conditions and agricultural practices.

Response 2: We agree that the composition of primary and secondary metabolites in plants is a result of many environmental biotic and abiotic factors. As the plants that we sampled in the research were not grown by us like one does in controlled experiment settings, there is a multitude of factors such as soil conditions, agricultural practices, and many others that could affect the metabolite profile of the samples. In the context of our present research, we aim to see whether there is a discernible pattern of metabolite profiles of the different P. betle variants in their natural settings (without us controlling specific factors that may affect the metabolite profile).

Other than that, the use of the term “variant” rather than “chemotypes” or “chemovars” reflect our caution in discriminating the samples as we admit that there were environmental factors that we did not measure or control or record in details in our research. This is the reason why we recorded the coordinates and altitude of the samples so that we can collect samples from the same locations for our next research, of which we plan to employ controlled experiment where we will also analyse the soil conditions and have proper record of the agricultural practices by farmers who grow the starting plant materials.

Point 3: In Table S1. In column “Growing condition” instead of “Grown under shade” should better had PAR measurements (photosynthetically active radiation). 

Response 3: Thank you for the suggestion and we will consider photosynthetically active radiation (PAR) measurements in our next research. “Grown under shade” is a generalized statement to describe the common traditional practice here in Malaysia to keep the P. betle plants healthy as P. betle is an outdoor shade-loving tropical plant and does not survive if grown under direct sunlight.       

Reviewer 2 Report

In my opinion the work reported is not very interesting but quite well done. Regarding the methodology, I have no comments.  I propose to accept the manuscript in the present form.

Author Response

Point 1: In my opinion, the work reported is not very interesting but quite well done. Regarding the methodology, I have no comments.  I propose to accept the manuscript in the present form.

Response 1: Thank you for the comment and we appreciate your review of the submitted manuscript.

Reviewer 3 Report

Review of manuscript plants-1424496 “Metabolomics-driven Discovery of an Introduced Species and Two Malaysian Piper betle L. Variants”

Summary
In this manuscript, the authors characterize different variants of Piper betle L. following a metabolomic approach based on NMR and GC-MS combined with chemometric data analysis tools. These tools allowed the discrimination of two known variants, “sirih melayu” and “sirih india” and the detection of samples of Piper rubro-venosum. Finally, the used approach identified some markers allowing the differentiation between these variants.
Commentaries
In my opinion, the manuscript is mainly well-written, and the research presented in this manuscript is sound. Furthermore, most of the experimental data support the results and conclusions presented by the authors. Therefore, I believe that this manuscript could be eventually published in Plants. However, some issues should be corrected by the authors to improve the clarity of the text and the reliability of the results.
* Introduction
I think that some references should be added related to previous works dealing with the metabolomic characterization of Piper betle variants. 
I do not feel that the last paragraph of the introduction is required. In my opinion, it could be moved to the methods section (or supplementary material – details of plant sampling) or, at least, strongly reduced.
* Results
- Section 2.1. Is it enough representative/reproducible to analyze six leaves from the same plant? Do the authors think that inherent variability between different samples will cause a dispersion of the samples in the scores space and a possible overlapping between variants?
- Please, give details on DNA barcoding (materials and methods).
* Methods
- NMR. Are there QC samples?
- Why were samples from different sources used for the NMR and GC-MS studies?
- What is the reason for using two GC-MS instruments?
- Line 642. Maybe just a single scaling approach is enough.
- Line 647. Please, give details on “default seven-groups cross-validation procedure was used”. As the number of samples is rather low, some information regarding how this CV approach performs is required.
- What about the PCA components larger than 2? Maybe there is also interesting information. In my opinion, the size of the models should be determined following usual numerical approaches.
Finally, some style and typographical errors should be corrected. A thorough language revision could benefit the manuscript.

Author Response

Point 1: In my opinion, the manuscript is mainly well-written, and the research presented in this manuscript is sound. Furthermore, most of the experimental data support the results and conclusions presented by the authors. Therefore, I believe that this manuscript could be eventually published in Plants. However, some issues should be corrected by the authors to improve the clarity of the text and the reliability of the results.

Response 1: Thank you for the comment and we appreciate your questions and suggestions, which truly help us improve the quality of the manuscript.

Point 2: *Introduction - I think that some references should be added related to previous works dealing with the metabolomic characterization of Piper betle variants. I do not feel that the last paragraph of the introduction is required. In my opinion, it could be moved to the methods section (or supplementary material – details of plant sampling) or, at least, strongly reduced.

Response 2: Thank you for the suggestion. Relevant references related to metabolomics of Piper betle variants have been added to the revised manuscript and the last paragraph of the Introduction section has been strongly reduced (page 2, lines 60–77). We have moved the details of plant sampling in the last paragraph of the Introduction section to the Methods section (page 11 (lines 461–462) and page 13 (lines 546–551)) and updated the numbering of references throughout the manuscript.

Point 3: *Results - Section 2.1. Is it enough representative/reproducible to analyse six leaves from the same plant? Do the authors think that inherent variability between different samples will cause a dispersion of the samples in the scores space and a possible overlapping between variants?  

Response 3: Thank you for the questions. We acknowledge the consequence of small sample size to the reproducibility of our research findings, hence the statement of the need for more samples in page 3, lines 144–150 to obtain more conclusive results. For this reason, we are transparent with the sample size by stating n = 1 instead of n = 6 for each sample. The pilot experiment involving 1H-NMR analysis of six leaves from one plant with three plants in total functions as a phase for us to rapidly identify the set of metabolite signals that are worth focusing on and the most appropriate analytical instrument in the subsequent phase of research. The results from the pilot experiment alone (without combination with GC-MS results) are not strong enough due to the sample size issue and this is complicated further by the inherent nature of 1H-NMR spectra (overlapping of metabolite signals), which hindered confident annotation of the signals to specific metabolite/s.

We do consider the probability of dispersion of the samples in the scores space and overlapping between variants. In the PCA of the pilot experiment, we assumed that at least two of the three plants end up having similar PC 1 and 2 scores, resulting in both plants cluster together in the scores space or at least, they are closer to each other than the remaining plant. However, the scores of “sirih merah” render this assumption unlikely as they suggest that it is not the same species as the other two plants. Aggravated by the small sample size issue, consequently it becomes difficult for us to confidently decide whether the other two plants belong to the same variant, or they truly are different variants. Despite these issues, the pilot experiment served its purpose of assisting us to select the set of metabolites (the phenylpropenes) and the most appropriate analytical instrument in the subsequent phase of research (GC-MS metabolomics), where we increased the sample size to n ≥ 6 for each P. betle variant.

Point 4: *Results - Please, give details on DNA barcoding (materials and methods).

Response 4: The details on how DNA barcoding analysis was performed have been added to the revised manuscript (page 12 (lines 523–540) and page 13 (lines 541–543)) and the numbering of references throughout the manuscript has been updated accordingly.

Point 5: *Methods - NMR. Are there QC samples?

Response 5: QC samples were analyzed in GC-MS metabolomics to assess the stability of the employed chromatography and mass spectrometry systems throughout the run sequence, especially in terms of retention time shift. As 1H-NMR spectral data acquisition involves neither chromatography nor mass spectrometry, we resolve to not having any QC sample in the analysis. Instead, D2O was set as the solvent lock to ensure consistency of 1H-NMR chemical shifts throughout the period of 1H-NMR spectral data acquisition.     

Point 6: *Methods - Why were samples from different sources used for the NMR and GC-MS studies?

Response 6: In our opinion, the use of same samples in both studies is not a good idea due to the issues of small sample size and nature of the replicates in the NMR study (not truly biological replicate as n = 1 for each variant). PCA of the NMR spectral data of 18 leaves suggested that we better focus our next analysis on the phenylpropenes, which are constituents of essential oil that are best analyzed using GC-MS. GC-MS metabolomics of essential oils necessitates the use of many leaves from one plant.

However, we could not fulfil this requirement using the same samples that were employed in the NMR study as there was only one plant for each variant at the location they were harvested. Hence, we searched for the locations where the different variants were grown via communication with the local community. This allowed us to increase the sample size from n = 1 to n = 6–8 for each P. betle variant in the GC-MS study. A statement on increasing sample size in the GC-MS study has been added to the revised manuscript (page 3, lines 144–150).         

Point 7: *Methods - What is the reason for using two GC-MS instruments?

Response 7: As we did not use any pure standards to identify the essential oil constituents, we applied two heuristic filters, namely similarity index > 80 when matched against the GC-MS spectral databases and linear retention index within ± 5 deviation from the literature data to reduce the number of misidentified constituents. The use of two GC-MS instruments in our research acts as the third filter, which we used to verify the identity of the constituents that were consistently detected by both GC-MS systems (page 6, lines 265–273). Using this third filter also showed the set of P. betle leaf essential oil constituents that at least shall be identified if our research is to be replicated in the future.

Point 8: *Methods - Line 642. Maybe just a single scaling approach is enough. 

Response 8: We beg to differ on the use of single scaling approach in our research as scaling does affect the covariance matrices of the data sets, thus the final PCA model. We have included the reference below in page 15, line 676 in the revised manuscript. We quote the following statements:

“Scale matters with PCA.”

“As with all statistical methods, PCA can be misused. The scaling of variables can cause different PCA results, and it is very important that the scaling is not adjusted to match prior knowledge of the data. If different scalings are tried, they should be described.”

(Source: Lever, J.; Krzywinski, M.; Altman, N. Points of significance: Principal component analysis. Nat. Methods 2017, 14, 641–642. https://doi.org/10.1038/nmeth.4346)

We performed mean-centering, UV, and Pareto scaling to demonstrate their effects on the resulting PCA score and loading plots, which were used to interpret the PCA model. Our aim was to use the one method that resulted in most intuitive loading plot for us to begin analysis of the variables starting with the ones that had the highest loadings in the PC 1 and 2 loadings space.

As shown in Figure S2a, the loading plot of mean-centered data matrix clearly shows high loadings of three variables (δ 1.92, 3.76, and 3.80) in the PC 1 and 2 loadings space. On the contrary, the loadings of the variables in the PC 1 and 2 loadings space are rather sparse for UV- and Pareto-scaled data matrix, making them less intuitive for us to pinpoint those variables with highest loadings. Besides, application of UV- and Pareto-scaling necessitates retention of more than two PCs in the final model to capture similar percentage of variation explained by two-PC model of the mean-centered data matrix.

On the other hand, the score plots in Figure S2 indicate that the distribution of sample scores in the scores space do not differ significantly with the application of the three preprocessing methods. This property implies that the PC 1 and 2 retain the same information regarding similarities between the samples in the scores space. From our previous experience, this is not the case for other data matrices that we did analyze.     

Point 9: *Methods - Line 647. Please, give details on “default seven-groups cross-validation procedure was used”. As the number of samples is rather low, some information regarding how this CV approach performs is required.

Response 9: It is stated in page 532 of SIMCA-P 14 software user guide (https://www.dynacentrix.com/telecharg/SimcaP/SIMCA14_User_Guide.pdf), SIMCA-P 14 uses the CV approach suggested by Eastment and Krzanowski in 1982 (EK approach) and a proprietary PC estimation in the cross-validation rounds, where by default the data are divided into seven CV groups. Using simulations, previous research showed that compared to other five CV approaches, EK approach performed poorly (success rate < 20%) in finding a suitable number of PC for the PCA model when the number of samples = 10 and the number of variables = 100. However, when applied to real UV-vis spectroscopic data with a data matrix of dimensions 31×201, the EK approach performed equally well as the other five CV approaches, where all the CV approaches showed decrease in Predicted Residual Sum of Squares (PRESS) for the first three PCs [82].

The seven-group cross-validation procedure that was used in the PCA of GC-MS analysis II data matrix of dimension 21×639 resulted in two statistically significant PCs. Analysis of two PCs is relevant to our research, which is exploratory in nature. We have added relevant references in page 15, lines 676 and 679 in the revised manuscript.

Point 10: What about the PCA components larger than 2? Maybe there is also interesting information. In my opinion, the size of the models should be determined following usual numerical approaches.

Response 10: We did analyze the scree plots and loadings of the third PC in the PCA of NMR and GC-MS studies. For the NMR data, seven PCs were found to be statistically significant, yet we limited the model size to only two PCs because in our opinion, the percentage of variation explained by the first two PCs (94.90%) is large enough for meaningful interpretation of the loadings. For the GC-MS analysis I data, four PCs were found to be statistically significant, and the third PC (explained 7.93% of the variation) captured the variation of monoterpenes in the samples, which are essential oil constituents that are generally more volatile than the sesquiterpenes.

In our opinion, the recognition of monoterpenes as chemical markers has inherent issues as they have highest tendency to vaporize during various stages of sample preparation, plant materials hydrodistillation, and essential oil storage. This was supported by the third PC of GC-MS analysis II data, which was found to be not statistically significant. For these reasons, we limit the size of all PCA models in our research to only two PCs, which was considered good enough for us to focus on interpretation of > 60% of the variation predicted by the models.

On top of that, we have deposited all the raw data generated in the present research in a public repository (page 16, lines 729–730) to allow other researchers who are interested in our research to have an unlimited access to the raw data and further analyze them. In case one is interested to explore the PCA components larger than 2, the raw data can be freely downloaded when we have made it public, which will be immediately after the publication of the present research.

Point 11: Finally, some style and typographical errors should be corrected. A thorough language revision could benefit the manuscript.

Response 11: We would like to thank you for the comment. Indeed, upon revision, we have noticed several style and typographical errors. We have gone through the manuscript line by line and made the necessary corrections. 

Round 2

Reviewer 3 Report

Review of manuscript plants-1424496 “Metabolomics-driven Discovery of an Introduced Species and Two Malaysian Piper betle L. Variants”

I appreciate the effort the authors made to improve the manuscript and to take into consideration the comments and suggestions.

Therefore, I consider that now the manuscript is suitable for publication. However, the authors should carefully proof-read the text to avoid typos and punctuation errors.